# Inhibition of Glutamate Release from Rat Cortical Nerve Terminals by Dehydrocorydaline, an Alkaloid from *Corydalis yanhusuo*

**DOI:** 10.3390/molecules27030960

**Published:** 2022-01-31

**Authors:** Tzu-Yu Lin, I-Yen Chen, Ming-Yi Lee, Cheng-Wei Lu, Kuan-Ming Chiu, Su-Jane Wang

**Affiliations:** 1Department of Anesthesiology, Far-Eastern Memorial Hospital, New Taipei City 22060, Taiwan; drlin1971@gmail.com (T.-Y.L.); taiwanmuggle@gmail.com (I.-Y.C.); drluchengwei@gmail.com (C.-W.L.); 2Department of Mechanical Engineering, Yuan Ze University, Taoyuan 32003, Taiwan; 3Division of Cardiovascular Surgery, Cardiovascular Center, Far-Eastern Memorial Hospital, New Taipei 22060, Taiwan; mingyi.lee@gmail.com; 4Department of Nursing, Asia Eastern University of Science and Technology, New Taipei City 22060, Taiwan; 5Department of Photonics Engineering, Yuan Ze University, Taoyuan 32003, Taiwan; 6School of Medicine, Fu Jen Catholic University, New Taipei City 24205, Taiwan; 7Research Center for Chinese Herbal Medicine, College of Human Ecology, Chang Gung University of Science and Technology, Taoyuan 33303, Taiwan

**Keywords:** dehydrocorydaline, glutamate release, Ca^2+^ influx, synaptosomes, neuroprotection, MAPK/ERK/synapsin I

## Abstract

Excessive release of glutamate induces excitotoxicity and causes neuronal damage in several neurodegenerative diseases. Natural products have emerged as potential neuroprotective agents for preventing and treating neurological disorders. Dehydrocorydaline (DHC), an active alkaloid compound isolated from *Corydalis yanhusuo*, possesses neuroprotective capacity. The present study investigated the effect of DHC on glutamate release using a rat brain cortical synaptosome model. Our results indicate that DHC inhibited 4-aminopyridine (4-AP)-evoked glutamate release and elevated intrasynaptosomal calcium levels. The inhibitory effect of DHC on 4-AP-evoked glutamate release was prevented in the presence of the vesicular transporter inhibitor bafilomycin A1 and the N- and P/Q-type Ca^2+^ channel blocker ω-conotoxin MVIIC but not the intracellular inhibitor of Ca^2+^ release dantrolene or the mitochondrial Na^+^/Ca^2+^ exchanger inhibitor CGP37157. Moreover, the inhibitory effect of DHC on evoked glutamate release was prevented by the mitogen-activated protein kinase (MAPK)/extracellular signal-regulated kinase (ERK) inhibitor PD98059. Western blotting data in synaptosomes also showed that DHC significantly decreased the level of ERK1/2 phosphorylation and synaptic vesicle-associated protein synapsin I, the main presynaptic target of ERK. Together, these results suggest that DHC inhibits presynaptic glutamate release from cerebrocortical synaptosomes by suppressing presynaptic voltage-dependent Ca^2+^ entry and the MAPK/ERK/synapsin I signaling pathway.

## 1. Introduction

In the mammalian brain, glutamate is the primary excitatory neurotransmitter and is involved in many neurological functions, such as learning, memory, long-term potentiation, and synaptic plasticity [1,2,3]. Controlling extracellular glutamate levels within a physiological range is crucial to maintaining normal neuronal transmission and viability. Excessive glutamate release and activation of glutamate receptors, leading to neuronal overstimulation and excitotoxicity, have been implicated in several neurological or neurodegenerative disorders (NDs), such as epilepsy, multiple sclerosis, Alzheimer’s disease (AD), Huntington’s disease (HD), amyotrophic lateral sclerosis (ALS), and Parkinson’s disease (PD) [3,4,5]. Consequently, reducing glutamate release from nerve terminals may be essential for neuroprotection.

Dehydrocorydaline (DHC), an active alkaloid compound isolated from *Corydalis yanhusuo,* was originally proven to reduce noradrenaline release from adrenergic nerve terminals [6]. Based on prior research, DHC exerts various biological properties, such as anticancer, anti-coronary heart disease, acetylcholinesterase inhibitory, antithrombotic, antigastrointestinal ulcer, antimicrobial, anti-inflammation, antiviral and antinociceptive effects [7,8,9,10]. Previous studies have demonstrated that DHC exerts protective effects on the cardiovascular system [11,12]. In addition, DHC decreases the production of proinflammatory cytokines and possesses anti-inflammatory effects. In addition, DHC can change the content of monoamines in the brain to process antidepressant-like effects [13]. However, the neuroprotective effects of DHC are still unknown.

The present work aimed to investigate the potential effects of DHC on the amount of released glutamate and characterize the molecular mechanism underlying the effect of DHC. Isolated rat cerebral cortex never terminals termed synaptosomes were used to study synaptic transmission. The purified presynaptic terminal is a well-established model and is capable of accumulating, storing, and releasing neurotransmitters. Therefore, the experiments were conducted with synaptosomes by investigating the effects and possible mechanisms of DHC on evoked glutamate release. The synaptosomal plasma membrane potential, intrasynaptosomal Ca^2+^ concentration ([Ca^2+^]i), and downstream modulation of voltage-dependent Ca^2+^ channels (VDCCs) were also monitored. In addition, as phosphorylation of various kinases regulates vesicle mobilization and glutamate release, this study also examined which type of protein kinase signaling pathway is involved in DHC-regulated glutamate release.

## 2. Results

### 2.1. DHC Inhibited Glutamate Release from Rat Brain Cortical Synaptosomes

To investigate the mechanisms involved in the neuroprotection of DHC (Figure 1A), we evaluated the release of glutamate in synaptosomes depolarized with the K^+^ channel blocker 4-AP, which destabilizes membrane potential and opens voltage-dependent Ca^2+^ channels (VDCCs). The addition of 4-AP (1 mM) to synaptosomes incubated in the presence of 1.2 mM CaCl_2_ resulted in a marked stimulation of glutamate release. Pretreatment with DHC (20 μM) significantly reduced glutamate release evoked by 4-AP depolarization compared with the control group (Figure 1B) = [t(205) = 15.71; *p* < 0.0001; 43.4 ± 1.9% inhibition]. 4-AP-evoked glutamate release was inhibited by DHC in a dose-dependent manner. The maximum inhibition (71.4 ± 1.6%) was observed when the compound was applied at 50 μM DHC. The concentration of DHC resulting in 50% inhibition (IC_50_) of 4-AP-evoked glutamate release was 20.8 μM (Figure 1C). Based on these data, a DHC concentration of 20 μM was used in the following tests.

### 2.2. The Decrease in Glutamate Release Induced by DHC Was Mediated via Calcium-Dependent Synaptic Vesicle Exocytosis

The effect of DHC on 4-AP-evoked glutamate release may be accounted for by two independent mechanisms: one that involves Ca^2+^-dependent exocytosis and one that involves Ca^2+^-independent release that occurred by reversal of the glutamate transporter [14,15]. A series of experiments was conducted to determine which mechanism is involved in the effect of DHC. In the absence of extracellular Ca^2+^ in the reaction solution, Ca^2+^-independent release of glutamate was measured by depolarizing synaptosomes with 1 mM 4-AP. The level of glutamate release was markedly reduced in a calcium-free medium containing the exogenous Ca^2+^ chelator EGTA (300 μM) (*p* < 0.0001). Compared with the control group, preincubation with DHC produced no increase in the 4-AP-evoked release of glutamate [t(6) = 1.52; P = 0.18; (Figure 2)]. This result suggests that the effects of DHC depend on extracellular Ca^2+^. dl-threo-β-benzyloxyaspartate (dl-TBOA), an inhibitor of sodium-dependent glutamate/aspartate transporters, inhibited the uptake of glutamate and subsequently increased glutamate release (*p* < 0.0001). In the presence of dl-TBOA, DHC continued to significantly reduce the 4-AP-evoked release of glutamate [t(6) = 7.34; *p* < 0.0005; (Figure 2)]. Furthermore, the level of 4-AP-evoked glutamate release was decreased by the vacuolar ATPase inhibitor bafilomycin A1, which blocked glutamate uptake into vesicles (*p* < 0.0001). In the presence of bafilomycin A1, DHC failed to produce a significant inhibition of 4-AP-evoked glutamate release [t(10) = 0.66; P = 0.52; (Figure 2)]. These data suggest that inhibition of 4-AP-evoked glutamate release by DHC is due to a decrease in synaptic vesicle exocytosis.

### 2.3. DHC Did Not Change Synaptosomal Excitability

To understand the mechanism responsible for the DHC-mediated inhibition of glutamate release, the synaptosomal plasma membrane potential under resting conditions and upon depolarization was detected by using the membrane potential-sensitive fluorescent probe 30,30,30-Dipropylthiadicarbocyanine iodide [DiSC3(5)]. Figure 3A shows that 4-AP led to a fast increase in DiSC3(5) fluorescence (25.1 ± 1.5 fluorescence units/5 min). Preincubation of synaptosomes with DHC (20 µM) did not significantly change the 4-AP-mediated increase in DiSC3(5) fluorescence (23.8 ± 0.8 fluorescence units/5 min; P = 0.85). This result indicates that the observed inhibition of glutamate release by DHC is not caused by a decrease in synaptosomal excitability because DHC did not alter 4-AP-mediated depolarization. Furthermore, the effect of DHC on glutamate release under basal or depolarized (15 mM KCl) conditions was evaluated. High external [K^+^]-induced depolarization elicits the Na^+^ channel-independent and Ca^2+^ channel-dependent release of glutamate. The addition of KCl evoked controlled glutamate release of 5.5 ± 0.4 nmol/mg/5 min, which was reduced to 3.0 ± 0.3 nmol/mg/5 min in the presence of 20 μM DHC (*p* < 0.0001; Figure 3B).

### 2.4. DHC Decreased the 4-AP-Induced Increase in Intrasynaptosomal Ca^2+^ Concentration

Subsequently, we investigated the effect of DHC on the intrasynaptosomal Ca^2+^ concentration ([Ca^2+^]i) in our model using the fluorescent Ca^2+^ indicator Fura-2-acetoxymethyl ester (Fura-2-AM) (Figure 4). An increase in [Ca^2+^]i was also observed after the addition of 1 mM 4-AP to depolarize synaptosomes. Notably, DHC significantly decreased the 4-AP-evoked increase in [Ca^2+^]i compared with the control levels (*p* < 0.0001; Figure 4). These data suggest that the effects of DHC on glutamate release were mediated by changes in [Ca^2+^]i.

### 2.5. The DHC Mechanism for Inhibiting Glutamate Release Possibly Involves N- and P/Q-type Ca^2+^ Channels

VDCCs have been proposed to play an important role in controlling Ca^2+^ influx and synaptic transmission [16,17]. Glutamate release from synaptic vesicles is triggered mainly by the entry of Ca^2+^ through the Ca_v_2.2 (N-type) and Ca_v_2.1 (P/Q-type) Ca^2+^ channels [18,19]. Therefore, we hypothesized that Ca^2+^ channel activities inhibit 4-AP-evoked glutamate release by DHC. To test this hypothesis, we treated synaptosomes with Ca^2+^ channel blockers. ω-conotoxin MVIIC (ω-con-MVIIC), a wide spectrum blocker of N- and P/Q-type Ca^2+^ channels, used at a concentration of 2 μM, led to a reduction in the glutamate release response to 4-AP depolarization (Figure 5, *p* < 0.0001).

DHC (20 μM) alone reduced 4-AP-evoked glutamate release (*p* < 0.0001). The decrease in 4-AP-evoked glutamate release caused by DHC in association with ω-con-MVIIC was completely prevented. No significant difference was observed between glutamate release after ω-con-MVIIC treatment and after the combined treatment of ω-con-MVIIC and DHC (P = 0.9). This result suggests that DHC-mediated inhibition of 4-AP-evoked glutamate release may be triggered by decreasing Ca^2+^ influx through N- and P/Q-type Ca^2+^ channels.

In addition to Ca^2+^ influx across the plasma membrane, Ca^2+^ released from intracellular organelles is also involved in glutamate release [20]. Dantrolene, an inhibitor blocking calcium release from intracellular storage in the endoplasmic reticulum, and 7-chloro-5-(2-chlorophenyl)-1,5-dihydro-4,1-benzothiazepin-2(3*H*)-one (CGP37157), a selective antagonist of the mitochondrial Na^+^/Ca^2+^ exchanger, were used to study the action of DHC. Dantrolene (10 μM) alone decreased the level of 4-AP (1 mM)-evoked glutamate release, while the application of DHC (20 μM) still effectively decreased the level of 4-AP-evoked glutamate release (*p* < 0.001, Figure 5). Similar results were obtained using 10 μM CGP37157, which blocked Ca^2+^ efflux from mitochondria (*p* < 0.01, Figure 5).

### 2.6. Involvement of ERK1/2 in the DHC-Mediated Inhibition of 4-AP-Evoked Glutamate Release

Extracellular signal-regulated protein kinases 1 and 2 (ERK1/2 or p44/p42), protein kinase C (PKC), and protein kinase A (PKA) have been found to play an important role in the regulation of glutamate release at the presynaptic level [21,22,23]. To investigate the role of these protein kinases in synaptosomal neurotransmitter release, we studied the effect of the commonly used ERK kinase inhibitor 2-(2-amino-3-methoxyphenyl)-4*H*-1-benzopyran-4-one (PD98059), PKA inhibitor *N*-[2-(pbromocinnamylamino) ethyl]-5-isoquinolinesulfonamide (H89), and PKC inhibitor bisindolylmaleimide I (GF109203X) on DHC-mediated inhibition of glutamate release in response to 4-AP (Figure 6). In the presence of H89 (100 μM) or GF109203X (10 μM), DHC (20 μM) could still decrease the level of 4-AP-evoked glutamate release. A significant difference was observed between the release after H89 or GF109203X alone and after H89 or GF109203X and DHC treatment (for H89 and DHC: *p* < 0.001; for GF109203X and DHC: *p* < 0.05). In contrast, the inhibitory effect of DHC (20 μM) on 4-AP-evoked glutamate release was prevented in the presence of PD98059. When PD98059 and DHC were applied simultaneously, the inhibition of glutamate release following 4-AP depolarization was not significantly different from that measured in the presence of PD98059 alone (*p* > 0.05).

Figure 7 shows the effect of DHC on the levels of phosphorylated ERK1/2 in synaptosomes. Depolarization of purified synaptosomes with 1 mM 4-AP significantly increased ERK1/2 phosphorylation (207.0 ± 18.7%; *p* < 0.003), and this effect was prevented by DHC (62.4 ± 6.5%; *p* < 0.001, Figure 7A, Appendix A). MAPK/ERK-dependent synapsin phosphorylation plays an important role in establishing functional synaptic connections [24]. Similar results were obtained by analyzing the phosphorylation of synaptic vesicle-associated protein synapsin I (Figure 7B, Appendix A). 4-AP (1 mM) increased the phosphorylation of synapsin I in the presence of external Ca^2+^ (144.3 ± 11.8%; *p* < 0.01), and this phenomenon was also reduced after DHC treatment (87.6 ± 19.8%; *p* < 0.05).

## 3. Discussion

In the present study, we aimed to investigate the effect of DHC on the 4-AP-evoked release of glutamate, intrasynaptosomal Ca^2+^ concentration, synaptosomal plasma membrane potential, VDCCs, and activation of protein kinases in rat brain synaptosomes. To our knowledge, this study presents the first examination of the effect of DHC on endogenous glutamate release at the presynaptic level. Several possible mechanisms for the DHC-mediated inhibition of glutamate release are discussed as follows.

The release of neurotransmitters from a nerve terminal is a dynamic process. On the presynaptic side, neurotransmitter release, as well as the release process itself, can be modulated at several voltage-gated ion (e.g., Na^+^, K^+^, Ca^+^) channels [25,26]. To clarify the mechanism responsible for the inhibition of Ca^2+^-dependent glutamate release mediated by DHC, nerve terminals (synaptosomes) prepared from the cerebral cortex of rats were isolated. DHC could modulate glutamate release in response to 4-AP by a number of mechanisms—first by altering the synaptosomal membrane potential and downstream modulating Ca^2+^ influx into the terminus and second by directly regulating the access of Ca^2+^ through VDCCs. The first possibility seems unlikely for the following reasons: (1) DHC significantly inhibited 4-AP- and KCl-evoked glutamate release. Both Na^+^ and Ca^2+^ channels are involved in glutamate release induced by 4-AP, while only Ca^2+^ channels are involved in KCl-evoked glutamate release [27]. Thus, the DHC-mediated inhibition of glutamate release is unlikely to be involved in Na^+^ channels. (2) The addition of DHC did not alter the resting membrane potential and had no significant effect on the synaptosomal plasma membrane potential depolarized by 4-AP. This finding indicates that K^+^ conductance does not play a role in DHC-mediated inhibition of glutamate release. (3) DHC interferes with the Ca^2+^-dependent (exocytotic) component of KCl-evoked glutamate release without interfering with the Ca^2+^-independent (transporter-mediated) component, which depends only on the membrane potential [28]. The inhibitory effect of DHC on 4-AP-evoked glutamate release was effectively abolished by the autophagy inhibitor bafilomycin A1, which blocks glutamate uptake into synaptic vesicles, and not by the glutamate transporter inhibitor dl-TBOA. Therefore, these results suggest that DHC affects the physiological pool of glutamate release by decreasing the Ca^2+^-dependent exocytotic component of glutamate release.

In synaptic terminals, both intracellular Ca^2+^ release and extracellular Ca^2+^ influx through plasma membrane VDCCs contribute jointly to the elevation of cytoplasmic Ca^2+^ levels, which are coupled with glutamate release [19,23]. Using the fluorescent calcium indicator Fura-2-AM, this study demonstrated that the 4-AP-evoked increase in [Ca^2+^]c was reduced by DHC, indicating the inhibitory effect of DHC on glutamate release by decreasing presynaptic Ca^2+^ influx through VDCC. We next examined whether Ca^2+^ entry specifically through N- and P/Q-type VDCCs is involved in the action of DHC using ω-con-MVIIC, which selectively targets N- and P/Q-type Ca^2+^ channels. Our observation shows that DHC-mediated inhibition of 4-AP-evoked glutamate release from synaptosomes was abolished by N- and P/Q-type Ca^2+^ channel blockers ω-con-MVIIC, suggesting that the inhibition of 4-AP-evoked glutamate release by DHC is controlled by Ca^2+^ entry through N- and P/Q-type Ca^2+^ channels. Furthermore, the inhibitory effect of DHC was insensitive to the ER ryanodine receptor inhibitor dantrolene as well as the mitochondrial Na^+^/Ca^2+^ exchange inhibitor CGP37157. These data suggest that intracellular Ca^2+^ stored in the endoplasmic reticulum (ER) and mitochondria was not involved in the inhibitory effect of DHC on 4-AP-evoked glutamate release. Although the mechanism of DHC on presynaptic Ca^2+^ channels is currently unknown, these data indicate that DHC-mediated inhibition of glutamate release is caused by a reduction in Ca^2+^ influx through N- and P/Q-type Ca^2+^ channels, which are coupled to glutamate exocytosis in nerve terminals.

Kinases, such as PKA, PKC, and MAPK/ERK, might regulate the function of presynaptic VDCCs and glutamate release [22,23]. This study suggests that the inhibitory effect of DHC on 4-AP-evoked glutamate release is mediated by the MAPK/ERK signaling pathway for the following reasons: (i) the DHC-mediated inhibition of glutamate release was abolished by the MAPK/ERK kinase inhibitor PD98059; (ii) the PKA inhibitor H89 and the PKC inhibitor GF109203X had no effect on release inhibition; and (iii) the 4-AP-induced phosphorylation level of ERK1/2 and synapsin I at ERK1/2-dependent phosphor sites 4/5 was significantly decreased by DHC. Presynaptic MAPK/ERK transmits extracellular signals to intracellular targets and plays a role in neurotransmitter exocytosis [22,29,30]. Depolarization-stimulated Ca^2+^ influx can activate MAPK/ERK and phosphorylate synapsin I at phosphorylated sites 4/5. Phosphorylation of synapsin I in response to stimuli triggers the release of synaptic vesicles from the actin cytoskeleton. Consequently, more vesicles are held close to the active zone for neurotransmitter exocytosis, which ultimately results in the release of glutamate from a storage vesicle into the synaptic cleft [29,31]. These results imply that the inhibition of glutamate release mediated by DHC involves the suppression of MAPK/ERK-dependent synapsin I phosphorylation and the consequently decreased availability of synaptic vesicles. The other synapsin proteins, synapsin II and synapsin III are reported to be phosphorylated by MAPK/ERK, suggesting that they may also play a role in the regulation of glutamate release in nerve terminals [32,33,34].

DHC can pass through the blood–brain barrier and possesses the potential to treat central nervous system diseases [35]. While DHC could be detected in various brain regions of rats, how DHC delivers to the brain remains to be further explored [36]. Although DHC has been shown to have antidepressant, analgesic, and antinociceptive effects and exert antioxidative and anti-inflammatory activities in animal models [9,13,37], the precise mechanism of the neuroprotective effect of DHC has not yet been fully elucidated. Previous research has already demonstrated that compounds with potent antioxidative and anti-inflammatory activities might offer benefits in neuroprotective effects, as free radical-induced oxidative damage to the brain has been regarded as an important cause of neuronal death in a variety of neurodegenerative disorders of the CNS [38,39]. In the present study, DHC was reported to attenuate glutamate release from nerve terminals, which may specifically explain a certain level of its neuroprotective mechanism. This is because the excess release of glutamate may lead to excitotoxicity in both acute and chronic insults, such as ischemic stroke and neurodegenerative disorders [3,40]. Neuroprotective agents, e.g., riluzole, lamotrigine, and lubeluzole, have decreased glutamatergic transmission before it becomes neurotoxic [41,42,43]. In our study, DHC inhibited depolarization-evoked glutamate release in a concentration-dependent manner, with concentrations ranging from 50 to 1 μM and IC_50_ values of 20.8 μM. Jin et al. showed that systemic administration of DHC at a dose of 1.5–3 mg/kg could produce an antidepressant effect in an animal model of depression [13]. Jin et al. showed that systemic administration of DHC at a dose of 1.5–3 mg/kg could produce an antidepressant effect in an animal model of depression [13]. Administering DHC (3–10 mg/kg) to animals results in an inhibition of the production of inflammatory mediators and produces antinociceptive effects in inflammatory pain models [9]. Furthermore, in vitro studies have shown that DHC, at 2.5–20 μM, exerts anti-inflammatory effects in cultured cells [44]. The present study is consistent with these reports, in which the inhibitory effects of DHC were observed in a concentration range of 5–50 μM.

Excitotoxicity caused by excess glutamate has been implicated in the pathogenesis of acute and chronic neurodegenerative disorders. Moreover, several studies have reported that reduced brain glutamate levels are an important strategy for neuroprotective actions. In our present study, DHC caused a potent inhibition on the release of glutamate evoked by 4-AP via suppressing presynaptic Ca^2+^ influx through the N- and P/Q-type Ca^2+^ channel. Furthermore, this inhibition may depend, at least in part, on the MAPK/ERK/synapsin I pathway suppression. However, the significance of our finding to in vivo clinical situations remains to be determined.

## 4. Materials and Methods

### 4.1. Animals and Ethics

The animal care and experimental procedures complied with the National Institutes of Health Guidelines for the Care and Use of Laboratory Animals and were approved by the Far Eastern Memorial Hospital (Animal Care and Utilization Committee, approval number 102-02-14-A; 103-02-25-A; IACUC-2021-FEMH-003, 10/2020). All efforts were made to reduce the number of animals and minimize their pain, suffering, and distress. Male Sprague–Dawley rats (weighing 150–200 g) were obtained from BioLASCO Taiwan Co., Ltd. (Taipei, Taiwan) and housed in the animal facility of the Far Eastern Memorial Hospital under controlled environmental conditions (temperature: 22 ± 1 °C, humidity: 50% and a 12-h light/dark cycle). All animals were provided free access to fresh water and food.

### 4.2. Preparation of Synaptosomes

Purified synaptosomes were prepared from the cerebral cortex of one animal and then purified from other tissue components using density gradient ultracentrifugation [45]. Briefly, the tissue was homogenized in 10 volumes of ice-cold buffer (0.32 M sucrose, pH 7.4) using a Potter-Elvehjem tissue grinder and centrifuged at 3000× *g* for 2 min to remove nuclei and debris. After centrifugation at 15,000× g for 10 min, the crude synaptosomal pellet was collected and resuspended in ice-cold homogenization buffer (320 mM sucrose, pH 7.4).

### 4.3. Percoll Gradient Purification

The supernatant was gently layered on a discontinuous Percoll density gradient (3, 10, and 23% *v*/*v* in Tris-buffered 0.32 M sucrose) and then centrifuged at 32,500× *g* for 7 min. The synaptosomal layer at the 10/23% interface was collected and washed with isotonic physiological buffer (20 mM HEPES buffer, pH 7.4, containing 140 mM NaCl, 5 mM KCl, 5 mM NaHCO_3_, 1 mM MgCl_2_, 1.2 mM Na_2_HPO_4_, and 10 mM glucose). This suspension was further centrifuged at 12,000× *g* for 10 min. This step is necessary to separate most of the Percoll from synaptosomal pellets. After purification, the synaptosomes were placed into an isotonic physiological buffer with 1% BSA and stored at 4 °C for no more than 3 h until required.

### 4.4. Measurements of Continuous Glutamate Release

The measurement of glutamate release was performed by using a modification of the continuous fluorimetric assay (PerkinElmer LS-55 fluorescence spectrometers, PerkinElmer, Inc., Waltham, MA, USA) [46]. Briefly, aliquots (0.5 mg of protein) of synaptosomal pellets were incubated with isotonic physiological buffer plus 16 μM bovine serum albumin (BSA) in a stirred and thermostated cuvette at 37 °C for 10 min. The synaptosomes were preincubated with 2 mM nicotinamide adenine dinucleotide phosphate (NADP^+^), 50 units/mL glutamate dehydrogenase, and 1.2 mM CaCl_2_ for 5 min before being subjected to depolarization with 1 mM 4-aminopyridine (4-AP; Sigma, St. Louis, MO, USA) or 15 mM KCl. Released glutamate was recorded by measuring the increase in fluorescence at 460 nm following an excitation wavelength of 360 nm. The fluorescence signal was calibrated by adding a standard of exogenous glutamate (5 nmol) to the reaction mixture at the end of each experiment. The released glutamate was calculated until the fluorescence reached balance (approximately 5 min). DHC was purchased from ChemFaces (Wuhan, Hubei, China). The compound was isolated from the tubers of Corydalis (purity ≥ 98%) and identified by ^1^H-NMR, HPLC (Appendix A). Dantrolene, bafilomycin A1, dl-TBOA, CGP37157, PD98059, GF109203X, H89, and ω-conotoxin MVIIC were obtained from Tocris Cookson (Bristol, UK). EGTA, sodium dodecyl sulfate (SDS), and all other chemical reagents were obtained from Sigma-Aldrich Co. (St. Louis, MO, USA).

### 4.5. Measurement of Synaptosomal Membrane Potential

Fluorometric measurements of synaptosomal plasma membrane potential using the voltage-sensitive cationic dye DiSC3(5) were performed using a PerkinElmer LS-55 spectrofluorometer equipped with 646 nm excitation and 674 nm emission filters. DiSC3(5) was purchased from Invitrogen (Carlsbad, CA, USA). Rat cerebrocortical synaptosomes were resuspended in isotonic physiological buffer and transferred to a continuously stirred quartz cuvette at 37 °C. DHC was added 5 min before the addition of 5 μM DiSC3(5). After an additional 3 min incubation, 1.2 mM CaCl_2_ was added to the cuvette before being subjected to depolarization with 1 mM 4-aminopyridine. A PerkinElmer LS-55 spectrofluorometer was used to monitor the fluorescence intensity at ex/em = 646/674 nm every 2 s. Cumulative data were analyzed using GraphPad Prism 8.4.3 (GraphPad, San Diego, CA, USA), and the results are expressed in fluorescence units.

### 4.6. Measurement of Intrasynaptosomal Free Ca^2+^ Concentration

Changes in intrasynaptosomal free Ca^2+^ concentration ([Ca^2+^]i) were measured with the fluorescent dye Fura-2-AM following the method described previously [29]. Fura-2-AM were obtained from Invitrogen (Carlsbad, CA, USA). Synaptosomes were incubated for 30 min at 37 °C with gentle shaking in isotonic physiological buffer containing 5 μM Fura-2-AM and 1.2 mM CaCl_2_. Fura-2-Ca fluorescence was determined from dual-wavelength measurements at an emission wavelength of 505 nm and excitation wavelengths of 340 and 380 nm on a PerkinElmer spectrofluorometer LS-55 (Waltham, MA, USA). Data were recorded at 4 s intervals, and the calcium concentration was calculated according to Grynkiewicz et al. (1985) [47].

### 4.7. Immunoblotting

Purified synaptosomes (2 mg protein/mL) were denatured with sample buffer (Pierce™ Lane Marker Reducing Sample Buffer, Invitrogen, Carlsbad, CA, USA), and the samples were heated for 5 min at 95 °C. The Pierce™ Lane Marker Reducing Sample Buffer, Halt™ protease, and phosphatase inhibitor single-use cocktail and Pierce™ BCA protein assay kit were obtained from Thermo Fisher Scientific (Waltham, MA, USA). Equal amounts of protein (10 μg) were separated by electrophoresis on a polyacrylamide gel (GoPAGE TGN Precast Gel, 4–15%, Hsinchu City, Taiwan) followed by transfer electrophoretically onto a polyvinylidene difluoride (PVDF) membrane. The membrane was then blocked using 5% skim milk in Tris-buffered saline (TBS-T, 137 mM NaCl, 20 mM Tris–HCl, 0.1% Tween 20, pH 7.6) for 1 h at room temperature. Incubation with the primary antibodies (anti-phospho ERK1/2 1:1000, anti-ERK1/2 1:1000, anti-phospho synapsin I 1:1000, and anti-synapsin I in TBS-T 1% skim milk) was performed overnight at 4 °C, followed by incubation with a secondary antibody (1:10000 in TBS-T 1% skim milk) for 1 h. Rabbit polyclonal antibodies directed against phospho ERK1/2, phospho synapsin I ERK 1/2, synapsin I, and β-actin were purchased from Cell Signaling Technology (Beverly, MA, USA). After extensive washing, the membranes were then subjected to Immobilon Western Chemiluminescent HRP Substrate according to the manufacturer’s manual. Goat anti-rabbit IgG, peroxidase-conjugated secondary antibody, and Immobilon Western Chemiluminescent HRP Substrate were obtained from Merck KGaA (Darmstadt, Germany).

### 4.8. Statistical Analysis

All data are presented as the mean ± SEM. Datasets were analyzed statistically by GraphPad Prism 8.4.3 software (GraphPad, San Diego, CA, USA). Statistical differences between the two groups were measured using the Student’s paired t-test. One-way analysis of variance (ANOVA) followed by Tukey’s post-hoc multiple comparisons test was used to determine significant differences among three or more groups. GraphPad Prism software was used to generate all graphs and perform the calculations and statistical analyses.

## 5. Conclusions

In conclusion, the results of this study demonstrate that the action of DHC was specific. The inhibitory effect of DHC on evoked glutamate release was abolished by extracellular chelation of Ca^2+^ ions with EGTA and by a vesicular transporter inhibitor. This effect was unresponsive to glutamate transporter inhibitors. DHC did not change synaptosomal excitability but reduced the depolarization-induced increase in [Ca^2+^]i. The mechanism by which DHC inhibits glutamate release involves N- and P/Q-type Ca^2+^ channels but not Ca^2+^ efflux from intracellular organelles. Furthermore, our results also demonstrate that suppression of the MAPK/ERK/synapsin I pathway plays a part in the inhibitory effect of DHC on evoked glutamate release. However, the relevance of our findings to in vivo clinical situations remains to be determined, and further examination is required. This finding provides further understanding of the DHC mechanism of action and suggests the potential use of DHC in the treatment of neurodegenerative diseases, especially in pathological situations where excessive glutamate release occurs.

## Figures and Tables

**Figure 1 molecules-27-00960-f001:**
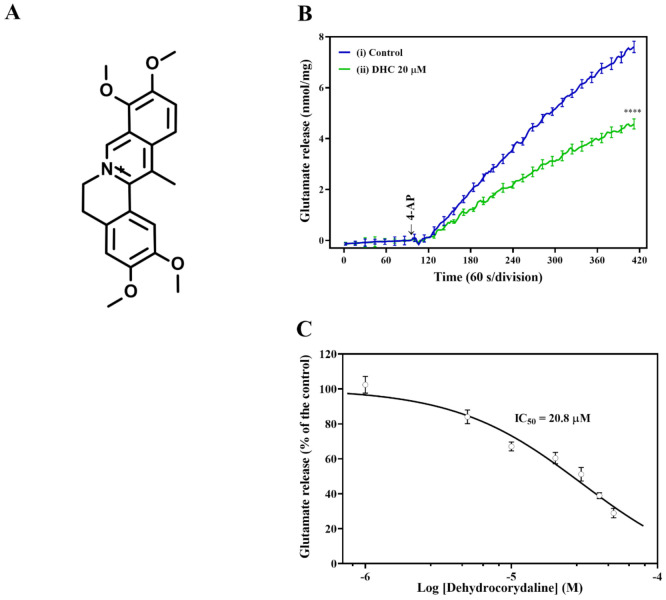
Influence of DHC on 4-AP-evoked glutamate release. (**A**) Chemical structure of DHC. (**B**) DHC (20 μM) inhibited glutamate release evoked by 4-AP depolarization. Rat brain cortical synaptosomes were depolarized with 15 mM KCl, and the continuous release of glutamate followed. Representative traces of continuous glutamate release curves after treatment with DHC in the presence of 1 mM 4-AP. (**C**) The dose-dependent response of DHC to glutamate release. Rat synaptosomes were preincubated for 10 min in the presence of DHC, followed by the addition of 4-AP. Data were analyzed with Prism 8.4.3 using nonlinear regression. The results are expressed as the mean ± SEM (*n* = 5). **** *p* < 0.0001 versus the control group.

**Figure 2 molecules-27-00960-f002:**
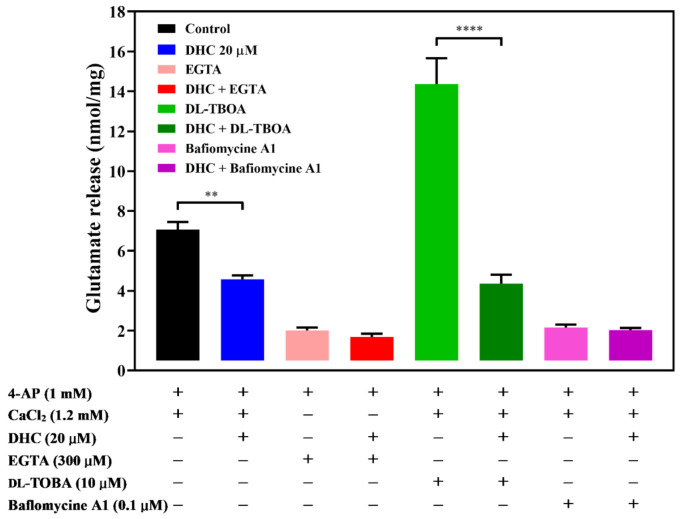
Effect of the Ca^2+^ chelator EGTA, glutamate transporter inhibitor dl-TBOA, or vacuolar ATPase inhibitor bafilomycin A1 on the action of DHC. Quantitative comparison of the extent of glutamate release by 1 mM 4-AP in the absence and presence of DHC. EGTA (300 μM), dl-TBOA (10 μM), bafilomycin A1 (0.1 μM) and DHC (20 μM) were added 10 min before depolarization. The results are the mean ± SEM values of independent experiments (*n* = 5). ** *p* < 0.01 versus the control group, **** *p* < 0.0001 versus the dl-TBOA-treated group.

**Figure 3 molecules-27-00960-f003:**
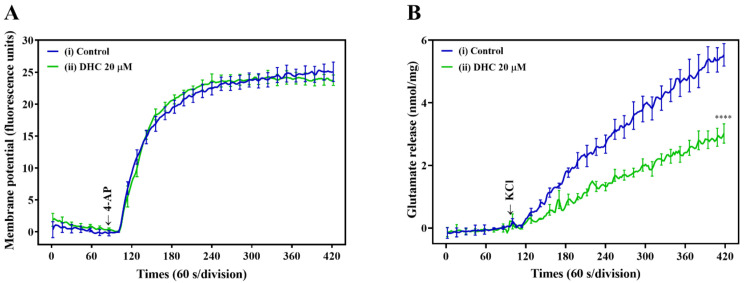
Influence of DHC on synaptosomal plasma membrane potential. (**A**) Response of the DiSC3(5) probe in synaptosome suspension to 4-AP depolarization. The arrow indicates the moment when depolarizing stimuli (4-AP 1 mM) were added to the synaptosome. The presented curve shows the results obtained in six independent measurements. (**B**) Synaptosomes were depolarized with 15 mM KCl, and the continuous release of glutamate followed. DHC (20 μM) decreased KCl-evoked glutamate release from synaptosomes. **** *p* < 0.0001 versus the control group (*n* = 5).

**Figure 4 molecules-27-00960-f004:**
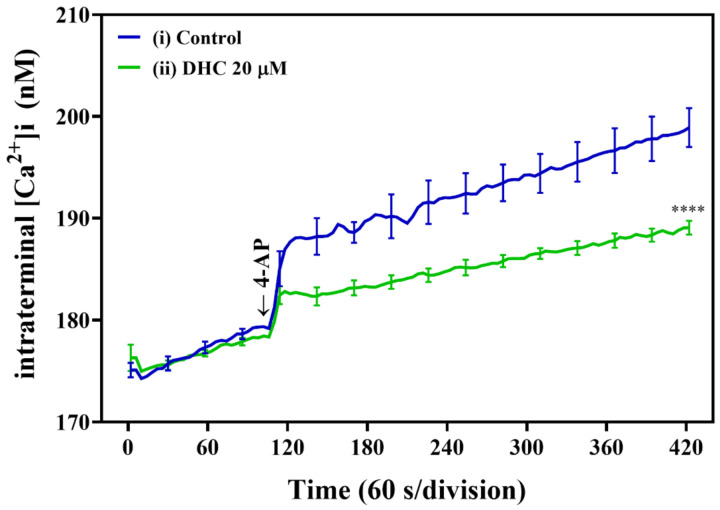
DHC reduced the 4-AP-induced increase in [Ca^2+^]i in mouse cortical brain synaptosomes. Synaptosomes were incubated with Fura-2 AM, followed by treatment with DHC (20 μM) as described. Representative ([Ca^2+^]i curves in the presence of DHC and 4-AP (1 mM) are plotted versus time. **** *p* < 0.0001 versus the control group (*n* = 5).

**Figure 5 molecules-27-00960-f005:**
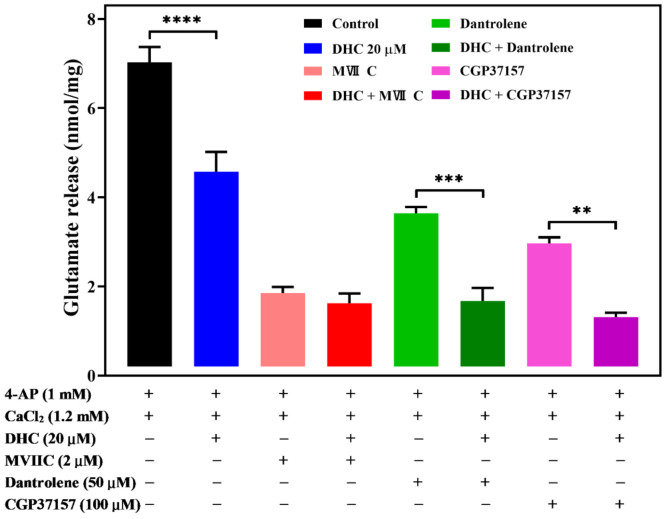
DHC inhibited 4-AP-evoked glutamate release through modulation of N- and P/Q- Ca^2+^ channels. Quantitative analysis of glutamate release by depolarizing stimuli (4-AP 1 mM) when synaptosomes were incubated with or without 20 μM DHC, 2 μM ω-con-MVIIC, 50 μM dantrolene, or 100 μM CGP37157. Data are the means ± SEM. **** *p* < 0.0001 versus the control group, *** *p* < 0.001 versus the dantrolene-treated group, ** *p* < 0.01 versus the CGP37157-treated group (*n* = 5).

**Figure 6 molecules-27-00960-f006:**
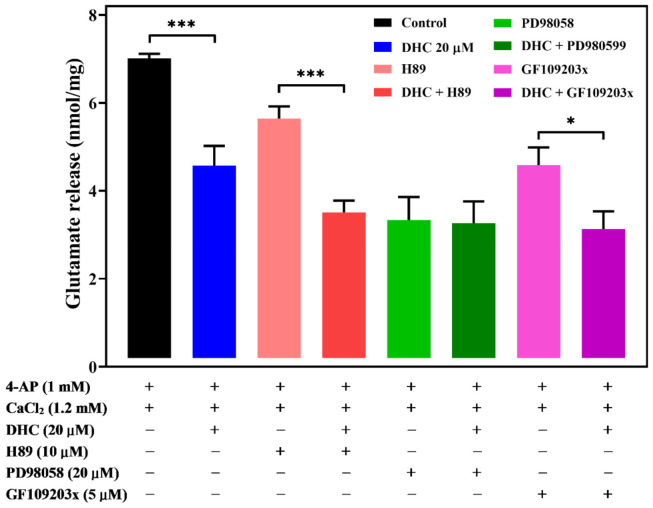
DHC-mediated inhibition of glutamate release is modulated by ERK1/2. Quantitative analysis of glutamate release by depolarizing stimuli (4-AP 1 mM) when synaptosomes were incubated with or without 20 μM DHC, 100 μM PKA inhibitor H89, 50 μM ERK kinase inhibitor PD98059, or 10 μM PKC inhibitor GF109203X. Data are the means ± SEM. *** *p* < 0.001 versus the control group or the H89-treated group, * *p* < 0.05 versus the GF109203X-treated group (*n* = 5).

**Figure 7 molecules-27-00960-f007:**
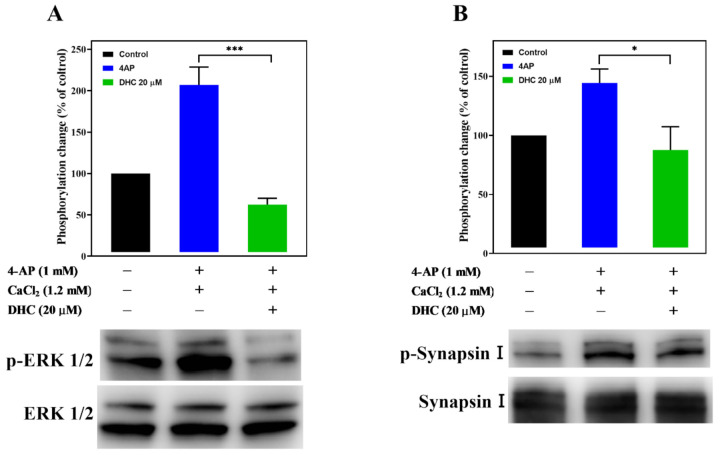
Effect of DHC on the activation of ERK1/2 and synapsin I phosphorylation. The expression levels of (**A**) p-ERK1/2, ERK1/2, (**B**) p-synapsin I, and synapsin I in synaptosomes were determined by immunoblotting. Data are the means ± SEM. For p-ERK1/2 expression, *** *p* < 0.001 versus the 4-AP alone group (*n* = 3). For synapsin I expression, * *p* < 0.05 versus the 4-AP alone group (*n* = 4).

## Data Availability

The data presented in this study are available on request from the corresponding author.

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
