# Peer review of "Inhibition of Glutamate Release from Rat Cortical Nerve Terminals by Dehydrocorydaline, an Alkaloid from Corydalis yanhusuo"

_molecules, 2022, doi:10.3390/molecules27030960_

Round 1

Reviewer 1 Report

As a general, the presented work is good and can be accepted if the next 2 main comments considered;

1-Title needs a full re-formulation.

2- In vivo testing of Dehydrocorydaline is required, at least to evaluate its ability to target brain cells.

Author Response

Response to reviewers

molecules-1556292

We thank the reviewer for the critical comments and constructive suggestions.

Response to reviewer1

As a general, the presented work is good and can be accepted if the next 2 main comments considered;

1-Title needs a full re-formulation.

Title has been modified to “Inhibition of glutamate release from rat cortical nerve terminals by dehydrocorydaline, an alkaloid from Corydalis yanhusuo” (Page 1, line 2-6)

2- In vivo testing of Dehydrocorydaline is required, at least to evaluate its ability to target brain cells.

We agree this point mentioned by the reviewer. In fact, in vivo animal studies will be our future experimental plan. Hope you can make allowances for this. Thank you very much for the suggestion of reviewer.

Reviewer 2 Report

The manuscript submitted by Lin et al and entitled "Dehydrocorydaline, an alkaloid from Corydalis yanhusuo, reduces depolarization-evoked glutamate release in rat brain cortical synaptosomes"  investigates the effect of dehydrocorydaline (DHC) on glutamate release using a rat brain cortical synaptosome model.

The biological results describe in this manuscript are original and correspond to the scope of the journal. However, the manuscript cannot be accepted in its present form since some experimental procedures are missing.

However, the author failed to give more details about the origin and quality of the natural product they used. Did they extract it by themself? If yes, which procedure did they use? How was the compound identified?

There is no standard deviation on Figure 7.

Author Response

Response to reviewers

molecules-1556292

We thank the reviewer for the critical comments and constructive suggestions.

Response to reviewer2

The manuscript submitted by Lin et al and entitled "Dehydrocorydaline, an alkaloid from Corydalis yanhusuo, reduces depolarization-evoked glutamate release in rat brain cortical synaptosomes" investigates the effect of dehydrocorydaline (DHC) on glutamate release using a rat brain cortical synaptosome model.

The biological results describe in this manuscript are original and correspond to the scope of the journal. However, the manuscript cannot be accepted in its present form since some experimental procedures are missing.

However, the author failed to give more details about the origin and quality of the natural product they used. Did they extract it by themself? If yes, which procedure did they use? How was the compound identified?

As suggestion by the reviewer, the sentence is modified to “DHC was purchased from ChemFaces (Wuhan, Hubei, China). The compound was isolated from the tubers of Corydalis (purity >=98%) and identified by 1H-NMR, HPLC.” (Page15, Line 389-392)

There is no standard deviation on Figure 7.

As suggestion by the reviewer, the standard error of the mean (SEM) are added (Page 9, line 216; Page 9, line 217; Page 9, line 220-222). However, the resulting ratios, expressed as percentage (%) change, are used to compare relative protein levels across the samples. All samples are compared to the control. Error bar representing negative control does not show because each of the data obtained from each sample is divided against the negative control which will give the output as percentage comparison with negative control.

Reviewer 3 Report

The manuscript Dehydrocorydaline, an alkaloid from Corydalis yanhusuo, reduces depolarization-evoked glutamate release in rat brain cortical synaptosomes is quite interesting and well designed however, few points required more attention by the authors.

Among these points:

1- It seems very strange that a quaternary alkaloids (charged) can be used to cross the blood brain barrier. The authors should mention clearly how can they deliver it ?

2- The lack of both positive and negative control in some biological experiments will limit its impact therefore the author are asked to check all the results for that

3-The graphs colors are very close It will be better to use different pattern 

4- The plots in figure 7 are heavily cropped, original image should be given in the supplementary 

5- In the discussion and conclusion, the mode of dehydrocorydaline action should be more discussed in relation to the major function groups present

Author Response

Response to reviewers

molecules-1556292

We thank the reviewer for the critical comments and constructive suggestions.

Response to reviewer3

The manuscript Dehydrocorydaline, an alkaloid from Corydalis yanhusuo, reduces depolarization-evoked glutamate release in rat brain cortical synaptosomes is quite interesting and welldesigned however, few points required more attention by the authors.

Among these points:

1- It seems very strange that a quaternary alkaloids (charged) can be used to cross the blood brain barrier. The authors should mention clearly how can they deliver it?

Our study was studied the effect of DHC on glutamate release by using isolated nerve terminals (synaptosomes).  Regarding this point, previous studies have shown that DHC could pass through blood–brain barrier. As suggestion by the reviewer, the sentence is added to the discussion section “While DHC could be detected in various brain regions of rats, how DHC delivers to the brain remains to be further explored” (Page 14, line 312-313). Follow two references are included in the discussion section.

Reference:

Gao, Y.; Hu, S.; Zhang, M.; Li, L.; Lin, Y. Simultaneous determination of four alkaloids in mice plasma and brain by LC-MS/MS for pharmacokinetic studies after administration of Corydalis Rhizoma and Yuanhu Zhitong extracts. J Pharm Biomed Anal 2014, 92, 6-12, doi:10.1016/j.jpba.2013.12.037.

Dou, Z.; Li, K.; Wang, P.; Cao, L. Effect of wine and vinegar processing of Rhizoma Corydalis on the tissue distribution of tetrahydropalmatine, protopine and dehydrocorydaline in rats. Molecules 2012, 17, 951-970, doi:10.3390/molecules17010951.

2- The lack of both positive and negative control in some biological experiments will limit its impact therefore the author are asked to check all the results for that

The aim of this study is to investigate the effect of DHC on 4-AP-evoked glutamate release. Thus, glutamate release evoked by 1mM 4-aminopyridine in the absence of DHC was used as control group in our experiment. In order to make the statement more clear, several graphs are modified. (Page 5, Line 119-120; Page 8, Line 190-191; Page 10, Line 224-225;Page 11, Line 231-232)

3-The graphs colors are very close It will be better to use different pattern 

As suggestion by the reviewer, the graphs colors are modified. (Page 5, Line 119-120; Page 8, Line 190-191; Page 10, Line 224-225;Page 11, Line 231-232)

4- The plots in figure 7 are heavily cropped, original image should be given in the supplementary 

As suggestion by the reviewer, original images of Western Blot are uploaded.

5- In the discussion and conclusion, the mode of dehydrocorydaline action should be more discussed in relation to the major function groups present

As suggestion by the reviewer,serveral sentences are added.“Excitotoxicity caused by excess glutamatehas been implicated in the pathogenesis of acute and chronic neurodegenerative disorders. Moreover, several studies have reported that reduced brain glutamate levels is an important strategy for neuroprotective actions. In our present study, DHC caused a potent inhibition on the release of glutamate evoked by 4-AP via suppressing presynaptic Ca2+ influx through N- and P/Q-type Ca2+ channel. Furthermore, this inhibition may depend, at least in part, on the MAPK/ERK/synapsin I pathway suppression. However, the significance of our finding to in vivo clinical situations remains to be determined.” (Page 14, Line 338-345)

Round 2

Reviewer 1 Report

Accept

Author Response

Response to reviewers

molecules-1556292

Response to reviewer1

Comments and Suggestions for Authors

Accept

We appreciate the reviewer’s acceptance of our paper for publication in the Molecules. We thank the reviewer for the critical comments and constructive suggestions.

Reviewer 2 Report

Please provide HPLC and HRMS or NMR profile of the compound as Supporting Information (or at least for review purposes).

Author Response

Response to reviewers

molecules-1556292

Response to reviewer2

Comments and Suggestions for Authors

Please provide HPLC and HRMS or NMR profile of the compound as Supporting Information (or at least for review purposes).

As suggestion by the reviewer, HPLC and NMR profile of the compound is uploaded.

Reviewer 3 Report

The manuscript has much improved and the authors amended most o the raised points

it could be accepted in the present form 

Author Response

Response to reviewers

molecules-1556292

Response to reviewer3

The manuscript has much improved and the authors amended most o the raised points

it could be accepted in the present form

We appreciate the reviewer’s acceptance of our paper for publication in the Molecules. We thank the reviewer for the critical comments and constructive suggestions.